# The Marginal Value of Adaptive Gradient Methods in Machine Learning

**Ashia C. Wilson[♯], Rebecca Roelofs[♯], Mitchell Stern[♯], Nathan Srebro[†], and Benjamin Recht[♯]**
{ashia,roelofs,mitchell}@berkeley.edu, nati@ttic.edu, brecht@berkeley.edu

[♯]University of California, Berkeley
[†]Toyota Technological Institute at Chicago

## Abstract

Adaptive optimization methods, which perform local optimization with a metric constructed from the history of iterates, are becoming increasingly popular for training deep neural networks. Examples include AdaGrad, RMSProp, and Adam. We show that for simple overparameterized problems, adaptive methods often find drastically different solutions than gradient descent (GD) or stochastic gradient descent (SGD). We construct an illustrative binary classification problem where the data is linearly separable, GD and SGD achieve zero test error, and AdaGrad, Adam, and RMSProp attain test errors arbitrarily close to half. We additionally study the empirical generalization capability of adaptive methods on several state-of-the-art deep learning models. We observe that the solutions found by adaptive methods generalize worse (often *significantly* worse) than SGD, even when these solutions have better training performance. These results suggest that practitioners should reconsider the use of adaptive methods to train neural networks.

## 1 Introduction

An increasing share of deep learning researchers are training their models with *adaptive gradient methods* [3, 12] due to their rapid training time [6]. Adam [8] in particular has become the default algorithm used across many deep learning frameworks. However, the generalization and out-of-sample behavior of such adaptive gradient methods remains poorly understood. Given that many passes over the data are needed to minimize the training objective, typical regret guarantees do not necessarily ensure that the found solutions will generalize [17].

Notably, when the number of parameters exceeds the number of data points, it is possible that the choice of algorithm can dramatically influence which model is learned [15]. Given two different minimizers of some optimization problem, what can we say about their relative ability to generalize? In this paper, we show that adaptive and non-adaptive optimization methods indeed find very different solutions with very different generalization properties. We provide a simple generative model for binary classification where the population is linearly separable (i.e., there exists a solution with large margin), but AdaGrad [3], RMSProp [21], and Adam converge to a solution that incorrectly classifies new data with probability arbitrarily close to half. On this same example, SGD finds a solution with zero error on new data. Our construction suggests that adaptive methods tend to give undue influence to spurious features that have no effect on out-of-sample generalization.

We additionally present numerical experiments demonstrating that adaptive methods generalize worse than their non-adaptive counterparts. Our experiments reveal three primary findings. First, with the same amount of hyperparameter tuning, SGD and SGD with momentum outperform adaptive methods on the development/test set across all evaluated models and tasks. This is true even when the adaptive methods achieve the *same training loss or lower* than non-adaptive methods. Second, adaptive methods often display faster initial progress on the training set, but their performance quickly

plateaus on the development/test set. Third, the same amount of tuning was required for all methods, including adaptive methods. This challenges the conventional wisdom that adaptive methods require less tuning. Moreover, as a useful guide to future practice, we propose a simple scheme for tuning learning rates and decays that performs well on all deep learning tasks we studied.

## 2   Background

The canonical optimization algorithms used to minimize risk are either stochastic gradient methods or stochastic momentum methods. Stochastic gradient methods can generally be written

$$w_{k+1} = w_k - \alpha_k \tilde{\nabla} f(w_k), \tag{2.1}$$

where $\tilde{\nabla} f(w_k) := \nabla f(w_k; x_{i_k})$ is the gradient of some loss function $f$ computed on a batch of data $x_{i_k}$.

Stochastic momentum methods are a second family of techniques that have been used to accelerate training. These methods can generally be written as

$$w_{k+1} = w_k - \alpha_k \tilde{\nabla} f(w_k + \gamma_k(w_k - w_{k-1})) + \beta_k(w_k - w_{k-1}). \tag{2.2}$$

The sequence of iterates (2.2) includes Polyak's heavy-ball method (HB) with $\gamma_k = 0$, and Nesterov's Accelerated Gradient method (NAG) [19] with $\gamma_k = \beta_k$.

Notable exceptions to the general formulations (2.1) and (2.2) are adaptive gradient and adaptive momentum methods, which choose a local distance measure constructed using the entire sequence of iterates $(w_1, \cdots, w_k)$. These methods (including AdaGrad [3], RMSProp [21], and Adam [8]) can generally be written as

$$w_{k+1} = w_k - \alpha_k H_k^{-1} \tilde{\nabla} f(w_k + \gamma_k(w_k - w_{k-1})) + \beta_k H_k^{-1} H_{k-1}(w_k - w_{k-1}), \tag{2.3}$$

where $H_k := H(w_1, \cdots, w_k)$ is a positive definite matrix. Though not necessary, the matrix $H_k$ is usually defined as

$$H_k = \text{diag}\left( \left\{ \sum_{i=1}^{k} \eta_i g_i \circ g_i \right\}^{1/2} \right), \tag{2.4}$$

where "$\circ$" denotes the entry-wise or Hadamard product, $g_k = \tilde{\nabla} f(w_k + \gamma_k(w_k - w_{k-1}))$, and $\eta_k$ is some set of coefficients specified for each algorithm. That is, $H_k$ is a diagonal matrix whose entries are the square roots of a linear combination of squares of past gradient components. We will use the fact that $H_k$ are defined in this fashion in the sequel. For the specific settings of the parameters for many of the algorithms used in deep learning, see Table 1. Adaptive methods attempt to adjust an algorithm to the geometry of the data. In contrast, stochastic gradient descent and related variants use the $\ell_2$ geometry inherent to the parameter space, and are equivalent to setting $H_k = I$ in the adaptive methods.

| | SGD | HB | NAG | AdaGrad | RMSProp | Adam |
|---|---|---|---|---|---|---|
| $G_k$ | $I$ | $I$ | $I$ | $G_{k-1} + D_k$ | $\beta_2 G_{k-1} + (1 - \beta_2) D_k$ | $\frac{\beta_2}{1 - \beta_2^k} G_{k-1} + \frac{(1 - \beta_2)}{1 - \beta_2^k} D_k$ |
| $\alpha_k$ | $\alpha$ | $\alpha$ | $\alpha$ | $\alpha$ | $\alpha$ | $\alpha \frac{1 - \beta_1}{1 - \beta_1^k}$ |
| $\beta_k$ | $0$ | $\beta$ | $\beta$ | $0$ | $0$ | $\frac{\beta_1(1 - \beta_1^{k-1})}{1 - \beta_1^k}$ |
| $\gamma$ | $0$ | $0$ | $\beta$ | $0$ | $0$ | $0$ |

**Table 1:** Parameter settings of algorithms used in deep learning. Here, $D_k = \text{diag}(g_k \circ g_k)$ and $G_k := H_k \circ H_k$. We omit the additional $\epsilon$ added to the adaptive methods, which is only needed to ensure non-singularity of the matrices $H_k$.

In this context, *generalization* refers to the performance of a solution $w$ on a broader population. Performance is often defined in terms of a different loss function than the function $f$ used in training. For example, in classification tasks, we typically define generalization in terms of classification error rather than cross-entropy.

## 2.1 Related Work

Understanding how optimization relates to generalization is a very active area of current machine learning research. Most of the seminal work in this area has focused on understanding how early stopping can act as implicit regularization [22]. In a similar vein, Ma and Belkin [10] have shown that gradient methods may not be able to find complex solutions at all in any reasonable amount of time. Hardt et al. [17] show that SGD is uniformly stable, and therefore solutions with low training error found quickly will generalize well. Similarly, using a stability argument, Raginsky et al. [16] have shown that Langevin dynamics can find solutions than generalize better than ordinary SGD in non-convex settings. Neyshabur, Srebro, and Tomioka [15] discuss how algorithmic choices can act as implicit regularizer. In a similar vein, Neyshabur, Salakhutdinov, and Srebro [14] show that a different algorithm, one which performs descent using a metric that is invariant to re-scaling of the parameters, can lead to solutions which sometimes generalize better than SGD. Our work supports the work of [14] by drawing connections between the metric used to perform local optimization and the ability of the training algorithm to find solutions that generalize. However, we focus primarily on the different generalization properties of adaptive and non-adaptive methods.

A similar line of inquiry has been pursued by Keskar et al. [7]. Hochreiter and Schmidhuber [4] showed that "sharp" minimizers generalize poorly, whereas "flat" minimizers generalize well. Keskar et al. empirically show that Adam converges to sharper minimizers when the batch size is increased. However, they observe that even with small batches, Adam does not find solutions whose performance matches state-of-the-art. In the current work, we aim to show that the choice of Adam as an optimizer itself strongly influences the set of minimizers that any batch size will ever see, and help explain why they were unable to find solutions that generalized particularly well.

## 3 The potential perils of adaptivity

The goal of this section is to illustrate the following observation: *when a problem has multiple global minima, different algorithms can find entirely different solutions when initialized from the same point.* In addition, we construct an example where adaptive gradient methods find a solution which has worse out-of-sample error than SGD.

To simplify the presentation, let us restrict our attention to the binary least-squares classification problem, where we can easily compute closed the closed form solution found by different methods. In least-squares classification, we aim to solve

$$\text{minimize}_w \quad R_S[w] := \tfrac{1}{2}\|Xw - y\|_2^2. \tag{3.1}$$

Here $X$ is an $n \times d$ matrix of features and $y$ is an $n$-dimensional vector of labels in $\{-1, 1\}$. We aim to find the best linear classifier $w$. Note that when $d > n$, if there is a minimizer with loss 0 then there is an infinite number of global minimizers. The question remains: what solution does an algorithm find and how well does it perform on unseen data?

### 3.1 Non-adaptive methods

Most common non-adaptive methods will find the same solution for the least squares objective (3.1). Any gradient or stochastic gradient of $R_S$ must lie in the span of the rows of $X$. Therefore, any method that is initialized in the row span of $X$ (say, for instance at $w = 0$) and uses only linear combinations of gradients, stochastic gradients, and previous iterates must also lie in the row span of $X$. The unique solution that lies in the row span of $X$ also happens to be the solution with minimum Euclidean norm. We thus denote $w^{\text{SGD}} = X^T(XX^T)^{-1}y$. Almost all non-adaptive methods like SGD, SGD with momentum, mini-batch SGD, gradient descent, Nesterov's method, and the conjugate gradient method will converge to this minimum norm solution. The minimum norm solutions have the largest *margin* out of all solutions of the equation $Xw = y$. Maximizing margin has a long and fruitful history in machine learning, and thus it is a pleasant surprise that gradient descent naturally finds a max-margin solution.

## 3.2 Adaptive methods

Next, we consider adaptive methods where $H_k$ is diagonal. While it is difficult to derive the general form of the solution, we can analyze special cases. Indeed, we can construct a variety of instances where adaptive methods converge to solutions with low $\ell_\infty$ norm rather than low $\ell_2$ norm.

For a vector $x \in \mathbb{R}^q$, let $\text{sign}(x)$ denote the function that maps each component of $x$ to its sign.

**Lemma 3.1** *Suppose there exists a scalar $c$ such that $X \, \text{sign}(X^T y) = cy$. Then, when initialized at $w_0 = 0$, AdaGrad, Adam, and RMSProp all converge to the unique solution $w \propto \text{sign}(X^T y)$.*

In other words, whenever there exists a solution of $Xw = y$ that is proportional to $\text{sign}(X^T y)$, this is precisely the solution to which all of the adaptive gradient methods converge.

**Proof** We prove this lemma by showing that the entire trajectory of the algorithm consists of iterates whose components have constant magnitude. In particular, we will show that

$$w_k = \lambda_k \, \text{sign}(X^T y) \,,$$

for some scalar $\lambda_k$. The initial point $w_0 = 0$ satisfies the assertion with $\lambda_0 = 0$.

Now, assume the assertion holds for all $k \le t$. Observe that

$$
\begin{aligned}
\nabla R_S(w_k + \gamma_k(w_k - w_{k-1})) &= X^T(X(w_k + \gamma_k(w_k - w_{k-1})) - y) \\
&= X^T \left\{ (\lambda_k + \gamma_k(\lambda_k - \lambda_{k-1}))X \, \text{sign}(X^T y) - y \right\} \\
&= \left\{ (\lambda_k + \gamma_k(\lambda_k - \lambda_{k-1}))c - 1 \right\} X^T y \\
&= \mu_k X^T y,
\end{aligned}
$$

where the last equation defines $\mu_k$. Hence, letting $g_k = \nabla R_S(w_k + \gamma_k(w_k - w_{k-1}))$, we also have

$$
H_k = \text{diag}\left( \left\{ \sum_{s=1}^k \eta_s \, g_s \circ g_s \right\}^{1/2} \right) = \text{diag}\left( \left\{ \sum_{s=1}^k \eta_s \mu_s^2 \right\}^{1/2} |X^T y| \right) = \nu_k \, \text{diag}\left( |X^T y| \right),
$$

where $|u|$ denotes the component-wise absolute value of a vector and the last equation defines $\nu_k$.

In sum,

$$
\begin{aligned}
w_{k+1} &= w_k - \alpha_k H_k^{-1} \nabla f(w_k + \gamma_k(w_k - w_{k-1})) + \beta_t H_k^{-1} H_{k-1}(w_k - w_{k-1}) \\
&= \left\{ \lambda_k - \frac{\alpha_k \mu_k}{\nu_k} + \frac{\beta_k \nu_{k-1}}{\nu_k}(\lambda_k - \lambda_{k-1}) \right\} \text{sign}(X^T y),
\end{aligned}
$$

proving the claim.[1] ∎

This solution is far simpler than the one obtained by gradient methods, and it would be surprising if such a simple solution would perform particularly well. We now turn to showing that such solutions can indeed generalize arbitrarily poorly.

## 3.3 Adaptivity can overfit

Lemma 3.1 allows us to construct a particularly pernicious generative model where AdaGrad fails to find a solution that generalizes. This example uses infinite dimensions to simplify bookkeeping, but one could take the dimensionality to be $6n$. Note that in deep learning, we often have a number of parameters equal to $25n$ or more [20], so this is not a particularly high dimensional example by contemporary standards. For $i = 1, \dots, n$, sample the label $y_i$ to be 1 with probability $p$ and $-1$ with probability $1 - p$ for some $p > 1/2$. Let $x_i$ be an infinite dimensional vector with entries

$$
x_{ij} = \begin{cases}
y_i & j = 1 \\
1 & j = 2, 3 \\
1 & j = 4 + 5(i-1), \dots, 4 + 5(i-1) + 2(1 - y_i) \\
0 & \text{otherwise}
\end{cases} .
$$

In other words, the first feature of $x_i$ is the class label. The next 2 features are always equal to 1. After this, there is a set of features *unique to $x_i$* that are equal to 1. If the class label is 1, then there is 1 such unique feature. If the class label is $-1$, then there are 5 such features. Note that the only discriminative feature useful for classifying data outside the training set is the first one! Indeed, one can perform perfect classification using only the first feature. The other features are all useless. Features 2 and 3 are constant, and each of the remaining features only appear for one example in the data set. However, as we will see, algorithms without such *a priori* knowledge may not be able to learn these distinctions.

Take $n$ samples and consider the AdaGrad solution for minimizing $\frac{1}{2}||Xw - y||^2$. First we show that the conditions of Lemma 3.1 hold. Let $b = \sum_{i=1}^{n} y_i$ and assume for the sake of simplicity that $b > 0$. This will happen with arbitrarily high probability for large enough $n$. Define $u = X^T y$ and observe that

$$
u_j = \begin{cases} n & j = 1 \\ b & j = 2,3 \\ y_j & \text{if } j > 3 \text{ and } x_{\lfloor \frac{j+1}{5} \rfloor, j} = 1 \\ 0 & \text{otherwise} \end{cases}
\quad \text{and} \quad
\text{sign}(u_j) = \begin{cases} 1 & j = 1 \\ 1 & j = 2,3 \\ y_j & \text{if } j > 3 \text{ and } x_{\lfloor \frac{j+1}{5} \rfloor, j} = 1 \\ 0 & \text{otherwise} \end{cases}
$$

Thus we have $\langle \text{sign}(u), x_i \rangle = y_i + 2 + y_i(3 - 2y_i) = 4y_i$ as desired. Hence, the AdaGrad solution $w^{\text{ada}} \propto \text{sign}(u)$. In particular, $w^{\text{ada}}$ has all of its components equal to $\pm \tau$ for some positive constant $\tau$. Now since $w^{\text{ada}}$ has the same sign pattern as $u$, the first three components of $w_{\text{ada}}$ are equal to each other. But for a new data point, $x^{\text{test}}$, the only features that are nonzero in both $x^{\text{test}}$ and $w^{\text{ada}}$ are the first three. In particular, we have

$$
\langle w^{\text{ada}}, x^{\text{test}} \rangle = \tau(y^{(test)} + 2) > 0 \,.
$$

Therefore, the AdaGrad solution will label all unseen data as a positive example!

Now, we turn to the minimum 2-norm solution. Let $\mathcal{P}$ and $\mathcal{N}$ denote the set of positive and negative examples respectively. Let $n_+ = |\mathcal{P}|$ and $n_- = |\mathcal{N}|$. Assuming $\alpha_i = \alpha_+$ when $y_i = 1$ and $\alpha_i = \alpha_-$ when $y_i = -1$, we have that the minimum norm solution will have the form $w^{\text{SGD}} = X^T \alpha = \sum_{i \in \mathcal{P}} \alpha_+ x_i + \sum_{j \in \mathcal{N}} \alpha_- x_j$. These scalars can be found by solving $XX^T \alpha = y$. In closed form we have

$$
\alpha_+ = \frac{4n_- + 3}{9n_+ + 3n_- + 8n_+ n_- + 3} \quad \text{and} \quad \alpha_- = \frac{4n_+ + 1}{9n_+ + 3n_- + 8n_+ n_- + 3} \,. \tag{3.2}
$$

The algebra required to compute these coefficients can be found in the Appendix. For a new data point, $x^{\text{test}}$, again the only features that are nonzero in both $x^{\text{test}}$ and $w^{\text{SGD}}$ are the first three. Thus we have

$$
\langle w^{\text{SGD}}, x^{\text{test}} \rangle = y^{\text{test}}(n_+ \alpha_+ - n_- \alpha_-) + 2(n_+ \alpha_+ + n_- \alpha_-) \,.
$$

Using (3.2), we see that whenever $n_+ > n_-/3$, the SGD solution makes no errors.

A formal construction of this example using a data-generating distribution can be found in Appendix C. Though this generative model was chosen to illustrate extreme behavior, it shares salient features with many common machine learning instances. There are a few frequent features, where some predictor based on them is a good predictor, though these might not be easy to identify from first inspection. Additionally, there are many other features which are sparse. On finite training data it looks like such features are good for prediction, since each such feature is discriminatory for a particular training example, but this is over-fitting and an artifact of having fewer training examples than features. Moreover, we will see shortly that adaptive methods typically generalize worse than their non-adaptive counterparts on real datasets.

## 4 Deep Learning Experiments

Having established that adaptive and non-adaptive methods can find different solutions in the convex setting, we now turn to an empirical study of deep neural networks to see whether we observe a similar discrepancy in generalization. We compare two non-adaptive methods – SGD and the heavy ball method (HB) – to three popular adaptive methods – AdaGrad, RMSProp and Adam. We study performance on four deep learning problems: **(C1)** the CIFAR-10 image classification task, **(L1)**

| Name | Network type | Architecture | Dataset | Framework |
|------|-------------|-------------|---------|-----------|
| C1 | Deep Convolutional | `cifar.torch` | CIFAR-10 | Torch |
| L1 | 2-Layer LSTM | `torch-rnn` | War & Peace | Torch |
| L2 | 2-Layer LSTM + Feedforward | `span-parser` | Penn Treebank | DyNet |
| L3 | 3-Layer LSTM | `emnlp2016` | Penn Treebank | Tensorflow |

**Table 2:** Summaries of the models we use for our experiments.[2]

character-level language modeling on the novel War and Peace, and (**L2**) discriminative parsing and (**L3**) generative parsing on Penn Treebank. In the interest of reproducibility, we use a network architecture for each problem that is either easily found online (C1, L1, L2, and L3) or produces state-of-the-art results (L2 and L3). Table 2 summarizes the setup for each application. We take care to make minimal changes to the architectures and their data pre-processing pipelines in order to best isolate the effect of each optimization algorithm.

We conduct each experiment 5 times from randomly initialized starting points, using the initialization scheme specified in each code repository. We allocate a pre-specified budget on the number of epochs used for training each model. When a development set was available, we chose the settings that achieved the best peak performance on the development set by the end of the fixed epoch budget. CIFAR-10 did not have an explicit development set, so we chose the settings that achieved the lowest training loss at the end of the fixed epoch budget.

Our experiments show the following primary findings: (*i*) Adaptive methods find solutions that generalize worse than those found by non-adaptive methods. (*ii*) Even when the adaptive methods achieve the *same training loss or lower* than non-adaptive methods, the development or test performance is worse. (*iii*) Adaptive methods often display faster initial progress on the training set, but their performance quickly plateaus on the development set. (*iv*) Though conventional wisdom suggests that Adam does not require tuning, we find that tuning the initial learning rate and decay scheme for Adam yields significant improvements over its default settings in all cases.

### 4.1 Hyperparameter Tuning

Optimization hyperparameters have a large influence on the quality of solutions found by optimization algorithms for deep neural networks. The algorithms under consideration have many hyperparameters: the initial step size $\alpha_0$, the step decay scheme, the momentum value $\beta_0$, the momentum schedule $\beta_k$, the smoothing term $\epsilon$, the initialization scheme for the gradient accumulator, and the parameter controlling how to combine gradient outer products, to name a few. A grid search on a large space of hyperparameters is infeasible even with substantial industrial resources, and we found that the parameters that impacted performance the most were the initial step size and the step decay scheme. We left the remaining parameters with their default settings. We describe the differences between the default settings of Torch, DyNet, and Tensorflow in Appendix B for completeness.

To tune the step sizes, we evaluated a logarithmically-spaced grid of five step sizes. If the best performance was ever at one of the extremes of the grid, we would try new grid points so that the best performance was contained in the middle of the parameters. For example, if we initially tried step sizes $2$, $1$, $0.5$, $0.25$, and $0.125$ and found that $2$ was the best performing, we would have tried the step size $4$ to see if performance was improved. If performance improved, we would have tried $8$ and so on. We list the initial step sizes we tried in Appendix D.

For step size decay, we explored two separate schemes, a development-based decay scheme (dev-decay) and a fixed frequency decay scheme (fixed-decay). For dev-decay, we keep track of the best validation performance so far, and at each epoch decay the learning rate by a constant factor $\delta$ if the model does not attain a new best value. For fixed-decay, we decay the learning rate by a constant factor $\delta$ every $k$ epochs. We recommend the dev-decay scheme when a development set is available;

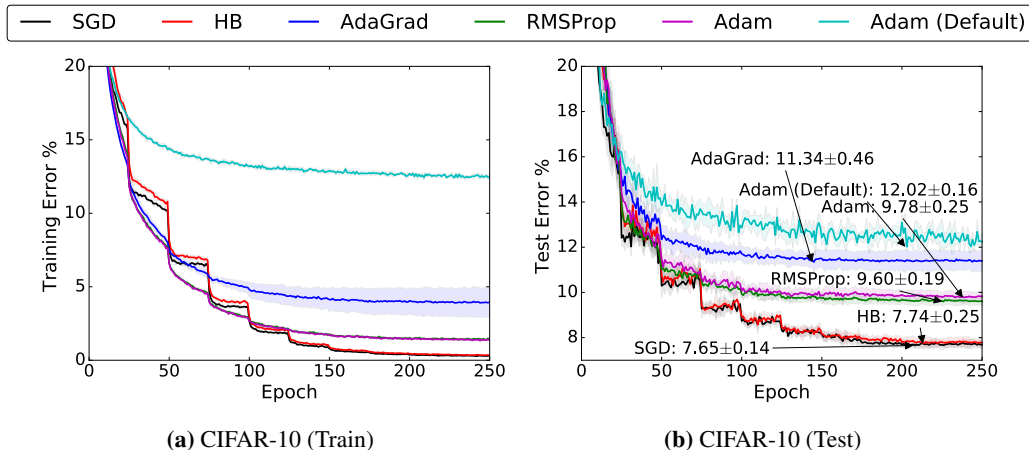

**Figure 1:** Training (left) and top-1 test error (right) on CIFAR-10. The annotations indicate where the best performance is attained for each method. The shading represents $\pm$ one standard deviation computed across five runs from random initial starting points. In all cases, adaptive methods are performing worse on both train and test than non-adaptive methods.

not only does it have fewer hyperparameters than the fixed frequency scheme, but our experiments also show that it produces results comparable to, or better than, the fixed-decay scheme.

## 4.2  Convolutional Neural Network

We used the VGG+BN+Dropout network for CIFAR-10 from the Torch blog [23], which in prior work achieves a baseline test error of $7.55\%$. Figure 1 shows the learning curve for each algorithm on both the training and test dataset.

We observe that the solutions found by SGD and HB do indeed generalize better than those found by adaptive methods. The best overall test error found by a non-adaptive algorithm, SGD, was $7.65 \pm 0.14\%$, whereas the best adaptive method, RMSProp, achieved a test error of $9.60 \pm 0.19\%$.

Early on in training, the adaptive methods appear to be performing better than the non-adaptive methods, but starting at epoch 50, even though the training error of the adaptive methods is still lower, SGD and HB begin to outperform adaptive methods on the test error. By epoch 100, the performance of SGD and HB surpass all adaptive methods on both train and test. Among all adaptive methods, AdaGrad's rate of improvement flatlines the earliest. We also found that by increasing the step size, we could drive the performance of the adaptive methods down in the first 50 or so epochs, but the aggressive step size made the flatlining behavior worse, and no step decay scheme could fix the behavior.

## 4.3  Character-Level Language Modeling

Using the `torch-rnn` library, we train a character-level language model on the text of the novel War and Peace, running for a fixed budget of 200 epochs. Our results are shown in Figures 2(a) and 2(b).

Under the fixed-decay scheme, the best configuration for all algorithms except AdaGrad was to decay relatively late with regards to the total number of epochs, either 60 or 80% through the total number of epochs and by a large amount, dividing the step size by 10. The dev-decay scheme paralleled (within the same standard deviation) the results of the exhaustive search over the decay frequency and amount; we report the curves from the fixed policy.

Overall, SGD achieved the lowest test loss at $1.212 \pm 0.001$. AdaGrad has fast initial progress, but flatlines. The adaptive methods appear more sensitive to the initialization scheme than non-adaptive methods, displaying a higher variance on both train and test. Surprisingly, RMSProp closely trails SGD on test loss, confirming that it is not impossible for adaptive methods to find solutions that generalize well. We note that there are step configurations for RMSProp that drive the training loss

below that of SGD, but these configurations cause erratic behavior on test, driving the test error of RMSProp above Adam.

### 4.4 Constituency Parsing

A constituency parser is used to predict the hierarchical structure of a sentence, breaking it down into nested clause-level, phrase-level, and word-level units. We carry out experiments using two state-of-the-art parsers: the stand-alone discriminative parser of Cross and Huang [2], and the generative reranking parser of Choe and Charniak [1]. In both cases, we use the dev-decay scheme with $\delta = 0.9$ for learning rate decay.

**Discriminative Model.**    Cross and Huang [2] develop a transition-based framework that reduces constituency parsing to a sequence prediction problem, giving a one-to-one correspondence between parse trees and sequences of structural and labeling actions. Using their code with the default settings, we trained for 50 epochs on the Penn Treebank [11], comparing labeled F1 scores on the training and development data over time. RMSProp was not implemented in the used version of DyNet, and we omit it from our experiments. Results are shown in Figures 2(c) and 2(d).

We find that SGD obtained the best overall performance on the development set, followed closely by HB and Adam, with AdaGrad trailing far behind. The default configuration of Adam without learning rate decay actually achieved the best overall training performance by the end of the run, but was notably worse than tuned Adam on the development set.

Interestingly, Adam achieved its best development F1 of 91.11 quite early, after just 6 epochs, whereas SGD took 18 epochs to reach this value and didn't reach its best F1 of 91.24 until epoch 31. On the other hand, Adam continued to improve on the training set well after its best development performance was obtained, while the peaks for SGD were more closely aligned.

**Generative Model.**    Choe and Charniak [1] show that constituency parsing can be cast as a language modeling problem, with trees being represented by their depth-first traversals. This formulation requires a separate base system to produce candidate parse trees, which are then rescored by the generative model. Using an adapted version of their code base,[3] we retrained their model for 100 epochs on the Penn Treebank. However, to reduce computational costs, we made two minor changes: (a) we used a smaller LSTM hidden dimension of 500 instead of 1500, finding that performance decreased only slightly; and (b) we accordingly lowered the dropout ratio from 0.7 to 0.5. Since they demonstrated a high correlation between perplexity (the exponential of the average loss) and labeled F1 on the development set, we explored the relation between training and development perplexity to avoid any conflation with the performance of a base parser.

Our results are shown in Figures 2(e) and 2(f). On development set performance, SGD and HB obtained the best perplexities, with SGD slightly ahead. Despite having one of the best performance curves on the training dataset, Adam achieves the worst development perplexities.

## 5   Conclusion

Despite the fact that our experimental evidence demonstrates that adaptive methods are not advantageous for machine learning, the Adam algorithm remains incredibly popular. We are not sure exactly as to why, but hope that our step-size tuning suggestions make it easier for practitioners to use standard stochastic gradient methods in their research. In our conversations with other researchers, we have surmised that adaptive gradient methods are particularly popular for training GANs [18, 5] and Q-learning with function approximation [13, 9]. Both of these applications stand out because they are not solving optimization problems. It is possible that the dynamics of Adam are accidentally well matched to these sorts of optimization-free iterative search procedures. It is also possible that carefully tuned stochastic gradient methods may work as well or better in both of these applications.

It is an exciting direction of future work to determine which of these possibilities is true and to understand better as to why.

## Acknowledgements

The authors would like to thank Pieter Abbeel, Moritz Hardt, Tomer Koren, Sergey Levine, Henry Milner, Yoram Singer, and Shivaram Venkataraman for many helpful comments and suggestions. RR is generously supported by DOE award AC02-05CH11231. MS and AW are supported by NSF Graduate Research Fellowships. NS is partially supported by NSF-IIS-13-02662 and NSF-IIS-15-46500, an Inter ICRI-RI award and a Google Faculty Award. BR is generously supported by NSF award CCF-1359814, ONR awards N00014-14-1-0024 and N00014-17-1-2191, the DARPA Fundamental Limits of Learning (Fun LoL) Program, a Sloan Research Fellowship, and a Google Faculty Award.

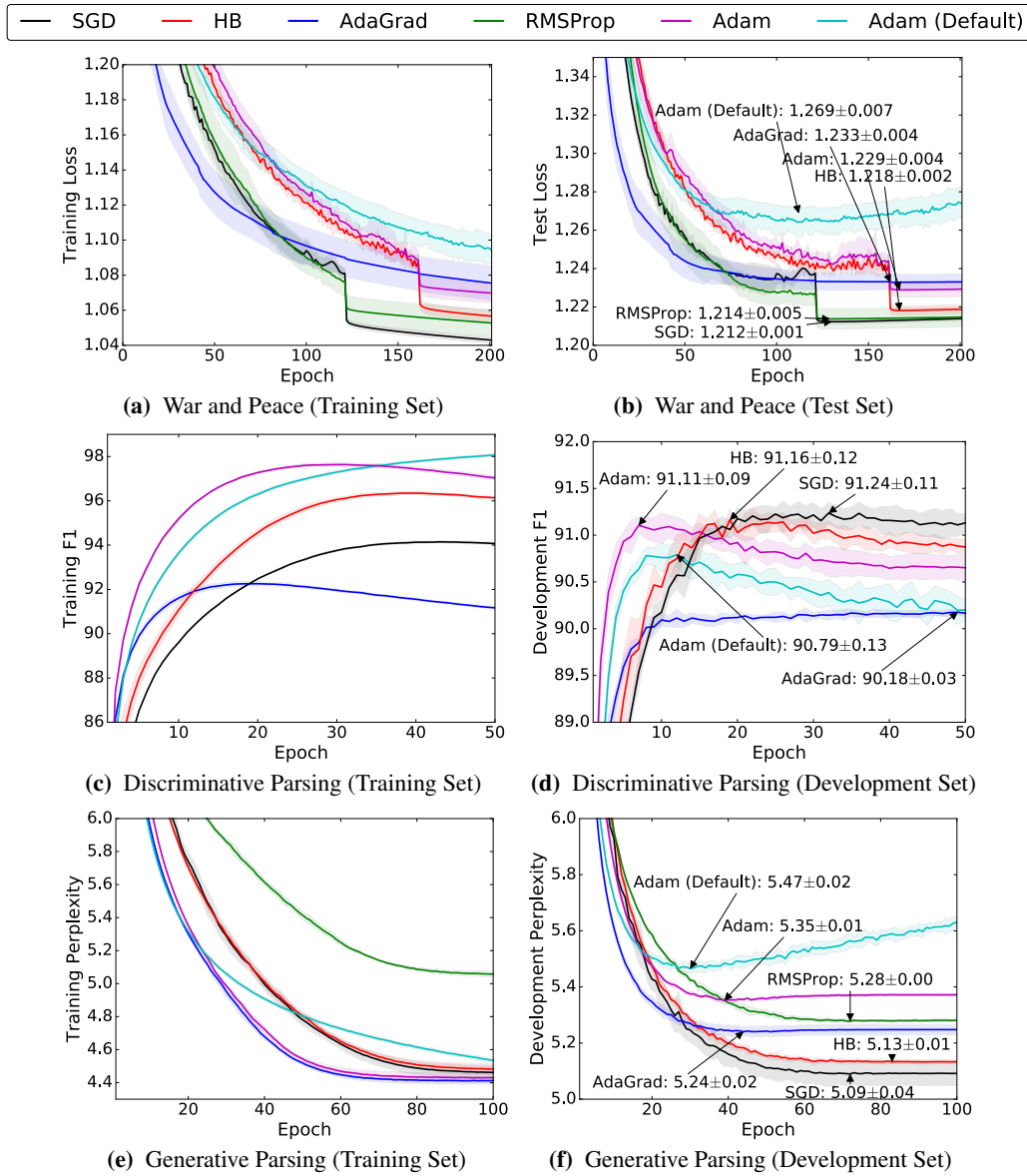

**Figure 2:** Performance curves on the training data (left) and the development/test data (right) for three experiments on natural language tasks. The annotations indicate where the best performance is attained for each method. The shading represents one standard deviation computed across five runs from random initial starting points.

## Footnotes

[1]In the event that $X^T y$ has a component equal to 0, we define $0/0 = 0$ so that the update is well-defined.

[2]Architectures can be found at the following links: (1) `cifar.torch`: `https://github.com/szagoruyko/cifar.torch`; (2) `torch-rnn`: `https://github.com/jcjohnson/torch-rnn`; (3) `span-parser`: `https://github.com/jhcross/span-parser`; (4) `emnlp2016`: `https://github.com/cdg720/emnlp2016`.

[3]While the code of Choe and Charniak treats the entire corpus as a single long example, relying on the network to reset itself upon encountering an end-of-sentence token, we use the more conventional approach of resetting the network for each example. This reduces training efficiency slightly when batches contain examples of different lengths, but removes a potential confounding factor from our experiments.

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
