[Supplementary Material]

# A  Full details of the minimum norm solution from Section 3.3

**Full Details.**  The simplest derivation of the minimum norm solution uses the kernel trick. We know that the optimal solution has the form $w^{\text{SGD}} = X^T \alpha$ where $\alpha = K^{-1} y$ and $K = XX^T$. Note that

$$
K_{ij} = \begin{cases}
4 & \text{if } i = j \text{ and } y_i = 1 \\
8 & \text{if } i = j \text{ and } y_i = -1 \\
3 & \text{if } i \neq j \text{ and } y_i y_j = 1 \\
1 & \text{if } i \neq j \text{ and } y_i y_j = -1
\end{cases}
$$

Positing that $\alpha_i = \alpha_+$ if $y_i = 1$ and $\alpha_i = \alpha_-$ if $y_i = -1$ leaves us with the equations

$$
(3n_+ + 1)\alpha_+ + n_- \alpha_- = 1,
$$
$$
n_+ \alpha_+ + (3n_- + 3)\alpha_- = -1.
$$

Solving this system of equations yields (3.2).

# B  Differences between Torch, DyNet, and Tensorflow

|  | Torch | Tensorflow | DyNet |
|---|---|---|---|
| SGD Momentum | 0 | No default | 0.9 |
| AdaGrad Initial Mean | 0 | 0.1 | 0 |
| AdaGrad $\epsilon$ | 1e-10 | Not used | 1e-20 |
| RMSProp Initial Mean | 0 | 1.0 | – |
| RMSProp $\beta$ | 0.99 | 0.9 | – |
| RMSProp $\epsilon$ | 1e-8 | 1e-10 | – |
| Adam $\beta_1$ | 0.9 | 0.9 | 0.9 |
| Adam $\beta_2$ | 0.999 | 0.999 | 0.999 |

**Table 3:** Default hyperparameters for algorithms in deep learning frameworks.

Table 3 lists the default values of the parameters for the various deep learning packages used in our experiments. In Torch, the Heavy Ball algorithm is callable simply by changing default momentum away from 0 with `nesterov=False`. In Tensorflow and DyNet, SGD with momentum is implemented separately from ordinary SGD. For our Heavy Ball experiments we use a constant momentum of $\beta = 0.9$.

# C  Data-generating distribution

We sketch here how the example from Section 3.3 can be modified to use a data-generating distribution. To start, let $\mathcal{D}$ be a uniform distribution over $N$ examples constructed as before, and let $D = \{(x_1, y_1), \ldots, (x_n, y_n)\}$ be a training set consisting of $n$ i.i.d. draws from $\mathcal{D}$. We will ultimately want to take $N$ to be large enough so that the probability of a repeated training example is small.

Let $\mathcal{E}$ be the event that there is a repeated training example. We have by a simple union bound that

$$
\mathbb{P}\left[\mathcal{E}\right] = \mathbb{P}\left[\bigcup_{i=1}^{n} \bigcup_{j=i+1}^{n} \{x_i = x_j\}\right] \leq \sum_{i=1}^{n} \sum_{j=i+1}^{n} \mathbb{P}\left[x_i = x_j\right] = \frac{n(n-1)}{2} \cdot \frac{1}{N} \leq \frac{n^2}{2N}.
$$

If the training set has no repeats, the result from Section 3.3 tells us that SGD will learn a perfect classifier, while AdaGrad will find a solution that correctly classifies the training examples but predicts $\hat{y} = 1$ for all unseen data points. Hence, conditioned on $\neg \mathcal{E}$, the error for SGD is

$$
\mathbb{P}_{(x,y) \sim \mathcal{D}}\left[\text{sign}\left\langle w^{\text{SGD}}, x \right\rangle \neq y \mid \neg \mathcal{E}\right] = 0,
$$

while the error for AdaGrad is

$$\mathbb{P}_{(x,y)\sim\mathcal{D}}\left[\operatorname{sign}\left\langle w^{\mathrm{ada}}, x\right\rangle \neq y \mid \neg\mathcal{E}\right]$$

$$= \mathbb{P}_{(x,y)\sim\mathcal{D}}\left[\operatorname{sign}\left\langle w^{\mathrm{ada}}, x\right\rangle \neq y \mid (x,y) \in D, \neg\mathcal{E}\right] \mathbb{P}_{(x,y)\sim\mathcal{D}}\left[(x,y) \in D \mid \neg\mathcal{E}\right]$$

$$+ \mathbb{P}_{(x,y)\sim\mathcal{D}}\left[\operatorname{sign}\left\langle w^{\mathrm{ada}}, x\right\rangle \neq y \mid (x,y) \notin D, \neg\mathcal{E}\right] \mathbb{P}_{(x,y)\sim\mathcal{D}}\left[(x,y) \notin D \mid \neg\mathcal{E}\right]$$

$$= 0 \cdot \frac{n}{N} + (1-p) \cdot \frac{N-n}{N}$$

$$= (1-p)\left(1 - \frac{n}{N}\right).$$

Otherwise, if there is a repeat, we have trivial bounds of 0 and 1 for the conditional error in each case:

$$0 \leq \mathbb{P}_{(x,y)\sim\mathcal{D}}\left[\operatorname{sign}\left\langle w^{\mathrm{SGD}}, x\right\rangle \neq y \mid \mathcal{E}\right] \leq 1,$$

$$0 \leq \mathbb{P}_{(x,y)\sim\mathcal{D}}\left[\operatorname{sign}\left\langle w^{\mathrm{ada}}, x\right\rangle \neq y \mid \mathcal{E}\right] \leq 1.$$

Putting these together, we find that the unconditional error for SGD is bounded above by

$$\mathbb{P}_{(x,y)\sim\mathcal{D}}\left[\operatorname{sign}\left\langle w^{\mathrm{SGD}}, x\right\rangle \neq y\right]$$

$$= \mathbb{P}_{(x,y)\sim\mathcal{D}}\left[\operatorname{sign}\left\langle w^{\mathrm{SGD}}, x\right\rangle \neq y \mid \neg\mathcal{E}\right]\mathbb{P}\left[\neg\mathcal{E}\right] + \mathbb{P}_{(x,y)\sim\mathcal{D}}\left[\operatorname{sign}\left\langle w^{\mathrm{SGD}}, x\right\rangle \neq y \mid \mathcal{E}\right]\mathbb{P}\left[\mathcal{E}\right]$$

$$= 0 \cdot \mathbb{P}\left[\neg\mathcal{E}\right] + \mathbb{P}_{(x,y)\sim\mathcal{D}}\left[\operatorname{sign}\left\langle w^{\mathrm{SGD}}, x\right\rangle \neq y \mid \mathcal{E}\right]\mathbb{P}\left[\mathcal{E}\right]$$

$$\leq 0 \cdot \mathbb{P}\left[\neg\mathcal{E}\right] + 1 \cdot \mathbb{P}\left[\mathcal{E}\right]$$

$$\leq \frac{n^2}{2N},$$

while the unconditional error for AdaGrad is bounded below by

$$\mathbb{P}_{(x,y)\sim\mathcal{D}}\left[\operatorname{sign}\left\langle w^{\mathrm{ada}}, x\right\rangle \neq y\right]$$

$$= \mathbb{P}_{(x,y)\sim\mathcal{D}}\left[\operatorname{sign}\left\langle w^{\mathrm{ada}}, x\right\rangle \neq y \mid \neg\mathcal{E}\right]\mathbb{P}\left[\neg\mathcal{E}\right] + \mathbb{P}_{(x,y)\sim\mathcal{D}}\left[\operatorname{sign}\left\langle w^{\mathrm{ada}}, x\right\rangle \neq y \mid \mathcal{E}\right]\mathbb{P}\left[\mathcal{E}\right]$$

$$= (1-p)\left(1 - \frac{n}{N}\right) \cdot \mathbb{P}\left[\neg\mathcal{E}\right] + \mathbb{P}_{(x,y)\sim\mathcal{D}}\left[\operatorname{sign}\left\langle w^{\mathrm{ada}}, x\right\rangle \neq y \mid \mathcal{E}\right]\mathbb{P}\left[\mathcal{E}\right]$$

$$\geq (1-p)\left(1 - \frac{n}{N}\right) \cdot \mathbb{P}\left[\neg\mathcal{E}\right] + 0 \cdot \mathbb{P}\left[\mathcal{E}\right]$$

$$\geq (1-p)\left(1 - \frac{n}{N}\right)\left(1 - \frac{n^2}{2N}\right).$$

Let $\epsilon > 0$ be a tolerance. For the error of SGD to be at most $\epsilon$, it suffices to take $N \geq \frac{n^2}{2\epsilon}$, in which case we have

$$\mathbb{P}_{(x,y)\sim\mathcal{D}}\left[\operatorname{sign}\left\langle w^{\mathrm{SGD}}, x\right\rangle \neq y\right] \leq \frac{n^2}{2N} \leq \epsilon.$$

For the error of AdaGrad to be at least $(1-p)(1-\epsilon)$, it suffices to take $N \geq \frac{n^2}{\epsilon}$ assuming $n \geq 2$, in which case we have

$$\mathbb{P}_{(x,y)\sim\mathcal{D}}\left[\operatorname{sign}\left\langle w^{\mathrm{ada}}, x\right\rangle \neq y\right] \geq (1-p)\left(1 - \frac{n}{N}\right)\left(1 - \frac{n^2}{2N}\right)$$

$$\geq (1-p)\left(1 - \frac{\epsilon}{n}\right)\left(1 - \frac{\epsilon}{2}\right)$$

$$\geq (1-p)\left(1 - \frac{\epsilon}{2}\right)\left(1 - \frac{\epsilon}{2}\right)$$

$$= (1-p)\left(1 - \epsilon + \frac{\epsilon^2}{4}\right)$$

$$\geq (1-p)(1-\epsilon).$$

Both of these conditions will be satisfied by taking $N \geq \max\left\{\frac{n^2}{2\epsilon}, \frac{n^2}{\epsilon}\right\} = \frac{n^2}{\epsilon}$.

Since the choice of $\epsilon$ was arbitrary, taking $\epsilon \to 0$ drives the SGD error to 0 and the AdaGrad error to $1 - p$, matching the original result in the non-i.i.d. setting.

# D Step sizes used for parameter tuning

## Cifar-10

- SGD: {2, 1, 0.5 (best), 0.25, 0.05, 0.01}
- HB: {2, 1, 0.5 (best), 0.25, 0.05, 0.01}
- AdaGrad: {0.1, 0.05, 0.01 (best, def.), 0.0075, 0.005}
- RMSProp: {0.005, 0.001, 0.0005, 0.0003 (best), 0.0001}
- Adam: {0.005, 0.001 (default), 0.0005, 0.0003 (best), 0.0001, 0.00005}

The default Torch step sizes for SGD (0.001) , HB (0.001), and RMSProp (0.01) were outside the range we tested.

## War & Peace

- SGD: {2, 1 (best), 0.5, 0.25, 0.125}
- HB: {2, 1 (best), 0.5, 0.25, 0.125}
- AdaGrad: {0.4, 0.2, 0.1, 0.05 (best), 0.025}
- RMSProp: {0.02, 0.01, 0.005, 0.0025, 0.00125, 0.000625, 0.0005 (best), 0.0001}
- Adam: {0.005, 0.0025, 0.00125, 0.000625 (best), 0.0003125, 0.00015625}

Under the fixed-decay scheme, we selected learning rate decay frequencies from the set $\{10, 20, 40, 80, 120, 160, \infty\}$ and learning rate decay amounts from the set $\{0.1, 0.5, 0.8, 0.9\}$.

## Discriminative Parsing

- SGD: {1.0, 0.5, 0.2, 0.1 (best), 0.05, 0.02, 0.01}
- HB: {1.0, 0.5, 0.2, 0.1, 0.05 (best), 0.02, 0.01, 0.005, 0.002}
- AdaGrad: {1.0, 0.5, 0.2, 0.1, 0.05, 0.02 (best), 0.01, 0.005, 0.002, 0.001, 0.0005, 0.0002, 0.0001}
- RMSProp: Not implemented in DyNet at the time of writing.
- Adam: {0.01, 0.005, 0.002 (best), 0.001 (default), 0.0005, 0.0002, 0.0001}

## Generative Parsing

- SGD: {1.0, 0.5 (best), 0.25, 0.1, 0.05, 0.025, 0.01}
- HB: {0.25, 0.1, 0.05, 0.02, 0.01 (best), 0.005, 0.002, 0.001}
- AdaGrad: {5.0, 2.5, 1.0, 0.5, 0.25 (best), 0.1, 0.05, 0.02, 0.01}
- RMSProp: {0.05, 0.02, 0.01, 0.005, 0.002 (best), 0.001, 0.0005, 0.0002, 0.0001}
- Adam: {0.005, 0.002, 0.001 (default), 0.0005 (best), 0.0002, 0.0001}