[Reviews · NeurIPS 2017]

Reviewer 1



// Summary: In an optimization and a learning context, the authors compare recently introduced adaptive gradient methods and more traditional gradient descent methods (with potential momentum). Adaptive methods are based on metrics which evolve along the optimization process. Contrary to what happens for gradient descent, Nesterov's method or the heavy ball method, this may result in estimates which are outside of the linear span of past visited points and estimated gradients. These methods became very popular recently in a deep learning context. The main question adressed by the authors is to compare both categories of method. First the authors construct an easy classification example for which they prove that adaptive methods behave very badly while non adaptive methods achieve perfect accuracy. Second the authors report extensive numerical comparisons of the different classes of algorithms showing consistent superiority of non adaptive methods. // Main comments: I do not consider myself knowledgable enough to question the relevance and correctness of the numerical section. This paper asks a very relevant question about adaptive methods. Giving a definitive answer is a complicated task and the authors provide elements illustrating that adaptive methods do not necessarily have a systematic advantage over non adaptive ones and that non adaptive optimization methods could be considered as serious challengers. The content of the paper is simple and really acessible, hilighting the simplicity of the question. Although simple, this remains extremely relevant and goes beyond the optimization litterature, questioning the reasons why certain methods reach very high popularity in certain communities. The elements given illustrate the message of the authors very clearly. The main points I would like to raise: - The artificial example given by the authors is only relevant for illustrative purposes. Comparison of algorithmic performances should be based on the ability to solve classes of problems rather than individual instances. Therefore, it is not really possible to draw any deep conclusion from this. For example, a bad choice of the starting point for non adaptive methods would make them equally bad as adaptive methods on this specific problem. I am not sure that the authors would consider this being a good argument against methods such as gradient descent. - Popularity of adaptive methods should be related to some practical advantage from a numerical point of view. I guess that positive results and comparison with non adaptive methods have already been reported in the litterature. Do the authors have a comment on this?

Reviewer 2



Summary This papers examines similarities and differences of adaptive and non-adaptive stochastic gradient methods in terms of their ability to generalize to unseen data. The authors find that adaptive methods generally generalize worse than non-adaptive ones both in handcrafted as well as in real world application scenarios. In addition to this, the authors give some advice as to how to tune non-adaptive methods and report that, despite being "adaptive", adaptive methods hugely benefit from tuning their hyperparameters. Comments Quality/Clarity This is a well done paper. The experiments presented confirm their theoretical implications and cover several popular practical applications. One question that comes up here though is, how general the applicability of the demonstrated behavior is on other applications and whether it also affects other adaptive methods such as natural stochastic gradient descent and its adaptive counterpart (natural stochastic gradient descent implicitly preconditions with the inverse of the Fisher information and its adaptive counterpart additionally steers the learning rate according to the empirical variance of the noisy gradients). Some minor issues: Eq. 6: one (probably the right one) should be \alpha_{-} l. 191 Optimized hyperparameters... or Optimization of hyperparameters has... Fig. 2 (a) and (b) How is it that errors drop for RMSProp and SGD on the one side and HB and Adam on the other at the very same moment? Why does this not apply to Adam (default) or AdaGrad? Further elaboration in the caption would be useful here. Originality/Significance The main finding, that adaptive methods fail to achieve better results in terms of test set error than non-adaptive ones, is of great use for practitioners and encourages the use of simpler, albeit more robust methods. The insight that adaptive methods are far from appropriately adapting to all hyperparameters should also be a hint when analyzing results of one such method.

Reviewer 3



The authors question the value of the adaptive gradient methods in machine learning. They claim that the adaptive methods like AdaGrad, RMSProp, Adam find different solutions than the non-adaptive variants like the Heavy Ball method or the Accelerated Gradient method. Their numerical experiments demonstrate that although the performances of the adaptive methods improve the objective function value quickly, particularly for the training set, their performances on test sets are worse than those of non-adaptive methods. Thus, the adaptive methods do not generalize well. To explain this poor performance of the adaptive methods, the authors construct a very smart and informative linear regression example and show that the adaptive method AdaGrad fails terribly for the unseen data, whereas the stochastic gradient descent method works as expected. The authors carefully setup a nice experimental study and demonstrate on a set of different machine learning problems that non-adaptive methods in general outperform the adaptive ones. This is a nice work. Especially the part in Section 3, where the authors construct their least-squares example. However, I am not sure whether their findings in this section may generalize to other machine learning problems including the ones in Section 4. Though, the empirical study does indeed show that adaptive methods perform worse than the non-adaptive methods, the link between Section 4 and Section 3 is not clear. I find it surprising that the practitioners have not observed the poor performances of the adaptive methods. The authors also share this feeling as they state in their conclusion. Overall, this work can be considered as a first step to look deeper into the generalization problems with the adaptive methods. Moreover, the practitioners can benefit from the nice empirical study.